# Egg Yolk as a New Source of Peptides with Antioxidant and Antimicrobial Properties

**DOI:** 10.3390/foods12183394

**Published:** 2023-09-11

**Authors:** Michał Czelej, Tomasz Czernecki, Katarzyna Garbacz, Jacek Wawrzykowski, Monika Jamioł, Katarzyna Michalak, Natalia Walczak, Agata Wilk, Adam Waśko

**Affiliations:** 1Biolive Innovation Sp. z.o.o., Dobrzańskiego 3, 20-262 Lublin, Poland; 2Department of Biotechnology, Microbiology and Human Nutrition, Faculty of Food Science and Biotechnology, University of Life Sciences in Lublin, Skromna 8, 20-704 Lublin, Poland; 3Department of Biochemistry, Faculty of Veterinary Medicine, University of Life Sciences in Lublin, Akademicka 12, 20-033 Lublin, Poland; 4Department of Epizootiology and Clinic of Infectious Diseases, Faculty of Veterinary Medicine, University of Life Sciences in Lublin, Gleboka 30, 20-612 Lublin, Poland

**Keywords:** antioxidation, egg yolk, enzymatic hydrolysis, bioactive peptides, proteins

## Abstract

A significant increase in interest in food-derived peptides obtained mostly through enzymatic reactions has been observed in the past few years. One of the best sources of bioactive peptides are defatted egg yolk proteins, which can potentially find application as high-quality nutritional supplements for infants with cow’s milk protein intolerance and as natural preservatives. The aim of this study was to obtain peptides from defatted egg yolk protein, to study their antioxidant and antimicrobial properties, and to identify peptides with bioactive properties To control the course of the process, MALDI-TOF/MS (matrix-assisted laser desorption/ionization time-of flight/mass spectrometry) spectra were also examined. The peptide mixture obtained through enzyme digestion was tested for its antioxidant properties by measuring the scavenging activity in 2,2-diphenyl-1-picrylhydrazyl radical (DPPH•), 2,2′-azinobis-(3-ethylbenzothiazoline-6-sulfonic acid) radical cation decolorization (ABTS•+), and ferric reducing activity (FRAP) assays. Antimicrobial activity was also studied. The peptide mixture exhibited significant antioxidant activity: DPPH—1776.66 ± 32.99, ABTS—390.43 ± 8.92, and FRAP—16.45 ± 0.19. The inhibition of bacterial growth by two concentrations of the peptide mixture was examined. The best result was obtained in *Bacillus cereus*, with an inhibition zone of 20.0 ± 1.0 and 10.7 ± 0.6 mm at the concentrations of 50 and 25 mg/mL, respectively. The results of the study suggest that the mixture of egg yolk peptides may exhibit a number of health-promoting properties.

## 1. Introduction

In recent years, society has become more conscious of the close relation between diet and health. This has resulted in the design of functional food that provides health benefits in addition to basic nutrition. Food and nutrition science has moved from identifying and correcting nutritional deficiencies to designing foods that promote optimal health and reduce the risk of diseases. Functional food is created thanks to a wide range of health-promoting additives. Bioactive peptides, which were described for the first time in 1979, might be one such additive [1]. Small peptides, especially di- and tripeptides, show high nutritional and therapeutic value, while larger ones (up to 50 amino acid residues and molecular masses less than 6000 Da) are presumed to be associated with the improved functionality of hydrolysates [2,3,4,5]. Bioactive peptides are inactive in the parent protein sequence, but during release by gastrointestinal digestion, fermentation, or enzymatic hydrolysis, they interact with receptors in the body and regulate the function of particular systems, e.g., nutrient uptake and immune defense. Furthermore, they take part in the transport of metal ions and opioids and have antioxidant, antimicrobial, antiproliferative, and ACE-inhibitory properties [6,7,8,9,10]. Their bioactive potency is dependent on the inherent composition and sequence of amino acid residues. 

There are several ways to obtain peptides. In the case of egg yolks, an important step preceding hydrolysis is defatting which is carried out to separate lipids, pigments, and aqueous and protein fractions. One of the most common methods of obtaining peptides is enzymatic proteolysis [6,11,12], which is considered to be highly important for the generation of bio-functional peptides. In comparison with chemical methods (acid or alkaline hydrolysis), it is characterized by mild reaction conditions, low amounts of undesirable products, and a high product yield and quality [5,13,14]. The hydrolysis mechanism generates free amino acid groups, which have, e.g., antioxidant properties and protect organisms and cells from oxidation damage. 

Butylated hydroxyanisole (BHA) and butylated hydroxytoluene (BHT) are cost-effective and efficient artificial antioxidants, but their toxicity and carcinogenicity raise concerns [15,16]. Given these reasons and their potential health benefits without side effects, natural antioxidants from food resources have become the object of widespread interest. It has also been shown that a large number of food-delivered peptides have antioxidant properties [3]. Peptides which exhibit antioxidative properties can be derived from different sources: animal (eggs, milk, whey) and plant (wheat, peanut kernel, rice, soybean, sunflower) proteins. One of the aforementioned possible sources of bioactive peptides that scientific research has focused on are hen eggs. Eggs are traditionally considered as a natural source of compounds with antioxidant properties, including several proteins, phospholipids, and certain micronutrients such as vitamin E, vitamin A, selenium, and carotenoids. Egg white is a commonly used source of protein and peptides. Egg yolk is mostly known from lecithin, which is widely used as a lubricant, surfactant, and emulsifying agent. Egg yolk can also provide lysozyme, cystatin, avidin, phosvitin, and phospholipids. Bioactive peptides from eggs include opioid peptides, peptides lowering high blood pressure, inhibiting platelet aggregation, and carrying metal ions, and peptides with immunostimulatory, antimicrobial, antidiabetic, and antioxidant activity [12,17,18]. It has been found that egg yolk proteins (EYPr) inhibit oxidation in a linoleate emulsion system [19]. Egg yolk phospholipids and phosvitin are reported to have antioxidant properties [20,21,22,23]. In vitro tests have shown a better digestibility of EYPr than those from milk casein treated with pepsin and trypsin [24]. 

The aim of this study was to obtain peptides from defatted egg yolk protein by enzymatic reaction and to determine the degree of hydrolysis (DH). Furthermore, the amino acid composition and the protein and fat content were evaluated. The research was complemented by antioxidant and antimicrobial studies and the identification of peptides with bioactive properties.

## 2. Materials and Methods

### 2.1. Materials

Liquid and pasteurized aseptically packed egg yolks taken from fresh class A eggs (Eipro, Lohne, Germany) were used in the experiment. One kilogram of the product is equal to 63 egg yolks. The product data specified by the manufacturer are presented in Table 1.

#### 2.1.1. Enzymes

Enzymatic hydrolysis was performed using commercially available pepsin from porcine gastric mucosa powder, ≥400 units/mg protein (P7125, Sigma-Aldrich, Saint Louis, MO, USA), and papain from papaya latex crude powder, 1.5–10 units/mg solid (P3375, Sigma-Aldrich, Saint Louis, MO, USA). 

#### 2.1.2. Other Reagents

The following compounds were used: ethanol (96%, POCH, Gliwice, Poland), deionized water (resistivity 18.2 MΩ × cm), sodium hydroxide (NaOH, POCH, Gliwice, Poland), trifluoroacetic acid (≥99.0%, Merck, Darmstadt, Germany), trichloroacetic acid (≥99.0%, Sigma-Aldrich, Saint Louis, MO, USA), 2,4,6-Tris(2-pyridyl)-s-triazine (≥99.0%, Sigma-Aldrich, Saint Louis, USA), FeCl_3_∙6H_2_O (97%,), sodium acetate trihydrate (≥99.0%, Merck, Germany), acetic acid (96%, Avantor, Gliwice, Poland), 1,1-diphenyl-2-picrylhydrazyl (97.0%, Sigma-Aldrich, Saint Louis, MO, USA), 2,2′-azino-bis(3-ethylbenzthiazoline-6-sulfonic acid) (98.0%, Roche, Basel, Switzerland), potassium persulfate (≥99.0%, Sigma-Aldrich, Saint Louis, MO, USA), phosphate-buffered saline (PBS, tablets, Sigma-Aldrich, Saint Louis, USA), barium hydroxide (95%, Sigma-Aldrich, Saint Louis, MO, USA), and hydrochloric acid (95%, POCH, Gliwice, Poland).

### 2.2. Methods

#### 2.2.1. Preparation of Egg Yolk Peptides

Defatted egg yolk proteins were obtained by ethanol (96%, PA, Chempur, Piekary Slaskie, Poland) extraction in mild conditions. The process was conducted in four steps (70% EtOH, 40% EtOH, 90% EtOH, water). In the first step, the egg yolks were vigorously mixed with 70% alcohol solutions in a ratio of 1: 1 [*v*:*v*]; next, the mixture was centrifuged at 3000× *g* for 30 min (MPW-380, MPW Med. Instruments, Warsaw, Poland). The liquid phase was removed and the sediment was used in the next step. The different steps were carried out analogously. In the last step carried out with water, the defatted egg yolk pellet was obtained and stored at −20 °C until further use. 

#### 2.2.2. Amino Acid Composition

In order to prepare the sample for the determination of non-sulfur amino acids, the acid hydrolysis of proteins was carried out in the presence of hydrochloric acid at an elevated temperature in accordance with the methodology proposed by Davies and Thomas [25]. Alkaline hydrolysis of proteins was carried out with barium hydroxide to determine the content of tryptophan. The protein sample for the determination of sulfur amino acids was prepared according to the procedure described by Schram and Moore [26]. It involves the oxidation of cysteine to cysteic acid and methionine to methionine sulfone, followed by hydrolysis in the presence of hydrochloric acid. Samples for tryptophan determination were centrifuged (15 min, 3000× *g*), and the other samples were evaporated and diluted. After filtration through a 0.22 μm filter, the samples were used for analysis. The amino acid determination was performed in an Ingos amino acid AAA 400 analyzer (Prague, Czech Republic) using low-ion ion-exchange chromatography with column derivatization (Ostion LG ANB) with ninhydrin and photometric detection at 570 nm and 440 nm for proline. The chromatographic separation was carried out in a system with four eluents, which consisted of buffers with pH 2.6, 3.0, 4.25, and 7.9.

#### 2.2.3. Protein Determination

In the experiment, the protein concentration was determined with the spectrophotometric method (MaestroNano Micro-Volume Spectrophotometer, MAESTROGEN, Hsinchu, Taiwan). Briefly, 2 µL of the protein sample was measured against water. The measurement was conducted at a 280-nm wavelength.

#### 2.2.4. Determination of Fat Content

The Soxhlet method was used to investigate the amount of lipids and the efficiency of the defatting process [27]. The procedure was carried out as follows: 1 g of lyophilized sample was placed in the extraction thimble and extracted for 4 h using hexane. After the extraction, the hexane was evaporated and the fat was placed in a dryer at 90 °C overnight. The next day, the flask was weighed and the fat content in the sample was determined.

#### 2.2.5. Enzymatic Reaction

The hydrolysis of the egg yolks was carried out as follows: (i) defatted egg yolk protein extract (111 g with 18% protein content) was dissolved in 1 L of MQ water; when the solution reached 70 °C, pH was adjusted to 6.0 using a 0.1 M NaOH solution; (ii) papain was added in a weight ratio of 10% relative to the EYPr mass; (iii) the mixture was stirred at 70 °C for 2 h; (iv) the enzymatic reaction was stopped by heating at 100 °C for 15 min; (v) the solution was cooled to 37 °C and the pH was adjusted to 3.0 using 1 M HCl; (vi) pepsin was added in a weight ratio of 10% relative to EYPr mass; (vii) the mixture was stirred at 37 °C for 2 h; (viii) the enzymatic reaction was stopped by heating at 100 °C for 15 min; (ix) the hydrolysate was cooled down and centrifuged at 3000× *g* for 30 min; (x) the supernatant was separated, lyophilized, and stored at 4 °C until further use.

#### 2.2.6. Degree of Hydrolysis (DH) 

The degree of hydrolysis was determined by the quantification of soluble peptides. For this purpose, 500 µL of the hydrolysate was collected in intervals (0, 0.5, 1, 1.5, and 2 h after enzyme addition) and mixed immediately with 500 µL of 10% trichloroacetic acid. After 15 min of incubation, the sample was centrifuged at 5500× *g* for 10 min. The content of soluble peptides in the supernatant was determined using the spectrophotometric method by measuring the absorbance at 280 nm. The degree of hydrolysis was calculated as the percentage of solubilized peptide to total protein content.

#### 2.2.7. Peptide Identification by MALDI-TOF/MS

MALDI TOF mass spectrometry is an appropriate method to check the presence and mass distribution of peptides in samples. This technique uses appropriate matrices, which cause the ionization and desorption of the tested material. Thanks to the time-of-flight analyzer measuring the time of passage of peptides to the detector, it is possible to determine the mass spectrum. 

MALDI spectra were collected to investigate the efficiency of the hydrolysis and to analyze the presence of lower MW peptides in the hydrolysate. A small amount of lyophilized hydrolysate powder was dissolved in 0.1% trifluoric acid and purified with Sample Prep Pipette Tips (ZipTip 0.2 μL C4 Millipore, Merck, Darmstadt, Germany) according to a standard procedure (Jehmlich, 2014) [28]. Next, 3 μL of purified peptide sample was mixed with 3 μL of an HCCA:DHA saturated matrix solution (50:50 [*v*:*v*]). The material was finally deposited on an AnchorChip MALDI plate with a hydrophobic coating (Bruker, Bremen, Germany). Mass spectra were recorded in the active positive reflection mode by an Ultraflex III MALDI TOF-TOF (Bruker, Bremen, Germany) spectrometer. All spectra were collected within the 500–5000 *m*/*z* range, smoothed (Savitzky-Golay method), and baseline corrected (Top Hat baseline algorithm) in flexAnalysis 3.0 software (Bruker, Bremen, Germany). The list of peaks with a signal-to-noise ratio greater than 3 was generated in the above-mentioned program. 

#### 2.2.8. Antioxidative Potential Estimation

To prepare peptide samples, 10 mg of each exactly weighed peptide lyophilizate sample was extracted in 1 mL of deionized water (5 min, 15 Hz, MM400, Retsch, Haan, Germany). After centrifugation (3 min, 15 °C, 33,000× *g*, 32R, Hettich, Tuttlingen, Germany), the supernatant was collected and used for further analyses. Absorbance measurements were carried out using a spectrophotometer (Marcel s330, Zielonka, Poland).

##### Ferric Reducing Antioxidant Power (FRAP) 

The ferric reducing ability of peptides was determined based on the FRAP method [29]. A 10 mmol/L solution of TPTZ (2,4,6-Tris(2-pyridyl)-s-triazine) in 40 mmol/L HCl was mixed with 20 mmol/L of FeCl_3_∙6H_2_O and 300 mmol/L of acetate buffer (pH 3.6) at a 1:1:10 *v*:*v*:*v* ratio. Then, 50 μL of the peptide solution was added to 150 μL of the mixture. The samples were mixed and incubated for 5 min at 25 °C. Absorbance was read at 593 nm against a blank sample. The Fe^2+^ ion concentration was assessed by referencing a standard curve established using known concentrations of FeSO_4_ solutions.

##### Determination of DPPH (1,1-diphenyl-2-picrylhydrazyl) Radical-Scavenging Activity

The free radical-scavenging activity measurements were conducted using the stable radical DPPH [30]. A total of 50 μL of the peptide solution was added to 150 μL of the DPPH• methanolic solution. The samples were incubated at 25 °C for 60 min on a horizontal shaker in darkness. After that, absorbance was read at 517 nm against a blank sample. The assessment of the antioxidant activity of the examined peptides relied on standard curves that had been created for Trolox and ascorbic acid equivalents.

##### Determination of ABTS•+ (2,2′-azino-bis(3-ethylbenzthiazoline-6-sulfonic acid) Radical Cation Decolorization Activity

The ABTS•+ working solution was prepared 16 h before measurements by mixing an equal volume of 7 mM ABTS and 2.4 mM potassium persulfate in water. Next, the working solution was diluted with PBS until absorbance reached 0.7 at 734 nm. A total of 1 mL of the peptide mixture sample was mixed with 50 µL of the working solution. Absorbance was read at 734 nm against a blank sample [31].

#### 2.2.9. Determination of Antibacterial Activity

In total, 10 strains of Gram (+) and Gram (−) bacteria belonging to different species were sampled from rotten fruits and vegetables. Species identification of the bacteria by MALDI-TOF/MS was carried out in accordance with the procedure described by Kosikowska et al. [32]. Bacteria were grown on Luria Bertani agar (Biocorp, Warsaw, Poland) at 30 °C for 24 h; then, they were subcultured until a homogenous cell culture was obtained. Colonies were picked and species identification using MALDI-TOF/MS (Bruker, Germany) was carried out. Ten species were identified: four belonging to the genera *Bacillus* (*B. cereus*, n = 2, *B. megaterium*, n = 1, *B. pumilus*, n = 1), *Kocuria rhizophila* (n = 1), *Serratia liquefaciens* (n = 1), *Pseudomonas aeruginosa* (n = 1), *Hafnia alvei* (n = 1), *Acinetobacter radioresistans* (n = 1), and *Stenotrophomonas maltophila* (n = 1).

The well diffusion test was performed using Mueller–Hinton agar (Biomaxima, Lublin, Poland). The inoculum was prepared using bacterial strains from a 24 h culture in Mueller–Hinton broth (Biomaxima, Lublin, Poland). The suspension of a 0.5 McFarland standard (1.5 × 10^8^) of each bacterial strain was made in a 0.85% sterile saline solution (Sigma-Aldrich, Saint Louis, MO, USA). The inoculated agar was poured into the assay plate. Next, four wells, each 7 mm in diameter, were cut out of the agar, and 50 μL of the EYPe (50 mg/mL) was placed into each well.

#### 2.2.10. Statistical Analysis 

Data are expressed as the mean ± standard deviation. One-way ANOVA followed by a post hoc Tukey’s test was preformed to analyze the significant differences between the data at the *p* value < 0.05 using Statistica Version 13.0 (StatSoft, Poland, TIBCO Software Inc., Palo Alto, CA, USA). 

## 3. Results & Discussion

### 3.1. Amino Acid Composition/Nitrogen Analysis

Table 2 presents the amino acid profiles in the egg yolk before and after the enzymatic hydrolysis. The amount of this bifunctional compound varies from 16.6 ± 0.6 (Trp) to 103.4 ± 5.7 mg/g (Glu). Notably, irrespective of the treatment, glutamic acid, aspartic acid, and leucine were the most abundant amino acids. These amino acids constituted a large portion of the egg yolk protein composition. A similar amino acid profile was obtained by Pokora et al. [33] in neutrase-digested hydrolysates; they also found high levels of serine and alanine. Comparable conclusions were reached by Attia et al. [34] and Adeyeye et al. [35], who determined high levels of these amino acids in egg yolks from different sources. Our study corroborates the findings reported by the aforementioned researchers. This agreement highlights the reliability of our findings, demonstrating a correlation between our results and the conclusions formulated by these authors. The increase in the quantities of almost every amino acid following the hydrolysis process is noteworthy. This may be attributed to the presence of enzymes, which can slightly increase the protein content in the mixture.

### 3.2. Determination of Fat Content by Soxhlet Extraction 

EYPr defatting is assumed to affect the protein solubility properties and limit their biological and biotechnological values (denaturation with ethanol or hexane [14,36,37]). It has been shown that controlled enzymatic hydrolysis is an effective method of protein modification, and this method has been used in various soy [38,39] and whey [40] proteins.

The average amount of lipids in the dry matter of egg yolk was equal to 50.6% ± 2.3, which is in agreement with the general state of knowledge. As shown by the results of the Soxhlet method, the extraction of lipids was not complete. After four steps of extraction, the content of fat decreased from 50.6% ± 2.3 to 18.9% ± 1.2 (Table 3). Therefore, the method is not as effective as lipid extraction with ethanol and hexane solutions (0.5%) [14]. While the lipid extraction efficiency may not reach its maximum level, the preservation of protein quality and the reduction of harmful conditions, including toxic reactants, are more prominent targets, potentially outweighing the associated limitations.

### 3.3. Degree of Hydrolysis DH

Enzymatic hydrolysis is the most applicable method for the extraction of peptides. The properties of this product depend to a great extent on the following parameters: enzyme (specificity and selectivity), temperature, pH, and enzyme-to-substrate ratio (E:S). In this experiment, papain and pepsin were used. The enzymes were selected based on previous preliminary studies on the DH of different enzymes. The temperature and pH applied during the experiment corresponded to the optimum for the enzymes, i.e., 70 °C, 6.0 and 37 °C, 3.0, respectively.

The DH is an important parameter describing enzymatic hydrolysis and may be responsible for controlling such factors as the composition and properties of peptides. The amount of enzyme can have a significant impact on the DH; therefore, the different amounts of the applied enzyme were examined (Table 4). The analysis of the effect of time and the enzyme concentration on hydrolysis efficiency revealed that there were statistically significant differences in the degree of hydrolysis obtained in samples with the 1:10 and 1:20 enzyme content relative to the amount of protein.

In each case, the biggest increase in DH was observed during the first 60 min of hydrolysis (Figure 1), beyond which the reaction progressed at a much slower rate and the DH changes were not significant. It can be suggested that the maximum peptide cleavage occurs during the first minute of hydrolysis. These results are in good agreement with those presented for other protein sources, e.g., fish [41] or wheat gluten [42]. Taking into consideration the results collected in Table 4, the best conditions for the enzymatic hydrolysis of defatted EYPr were the 10% enzyme addition (1:10 enzyme to substrate ratio). In this case, the DH increased from the intact value of 2.32% ± 0.04 to 34.71% ± 0.05. A remarkable outcome was attained through the implementation of a sequential two-step hydrolysis approach, comprising an initial 2 h papain hydrolysis, followed by a subsequent 2 h pepsin hydrolysis. Table 5 shows a comparison of the present results with those reported for a short digestion time (2–4 h) with such enzymes as neutrase, pronase, alcalase, or trypsin (25–33%) [33,37,43,44].

The employment of the two-step hydrolysis approach yielded significant DH improvement. The comparative analysis with other enzyme-based hydrolysis methods revealed the competitiveness of the presented approach in achieving a substantial DH increase.

### 3.4. Investigation of the Peptide Profile by MALDI-TOF/MS Analysis

In this research, we did not use any peptide standards in the analysis; hence, the discussion is limited to the presence or absence of special peaks in *m*/*z* ranges. After the two-step hydrolysis (DH 34.76%), the majority of the peptide peaks ranged from 700 to 1400 Da (Figure 2). In comparison with the peptide profile before the enzymatic hydrolysis, the distribution of the peaks changed so that a greater number of masses between 700 and 1100 Da were observed. High-molecular-weight peptides from the substrate were converted to smaller peptides with the progress of the reaction time. In the case of the raw egg yolk sample spectrum, there was a higher amount of heavier peptides and small proteins, which were cleaved after the digestion. The main conclusion is that the amount of peaks representing smaller MW peptides increased after the hydrolysis at the expense of larger masses. 

Based on the obtained spectrum of the hydrolysate sample and the *m*/*z* ratio, fragments of peptide sequences were determined by RapidDeNovo sequencing. The fragments were checked in the BIOPEP peptide activity database [46]. The final peptide sequences were determined using BLAST and a peptide mass calculator. 

Some of the peptides were assigned biological properties (Table 6). Among them, the peptides DSYEHGGEP, LMSYMWSTSM, PGVTYPHPGQDTSAG, and PTDQKVGWGGEGQIQ were identified as antioxidant compounds, and the peptides DSYEHGGEP, HVDLDEVANKIA, and FEDPERQESSRKE were identified as antimicrobial substances. These results strongly suggest the need for further investigation of the antioxidant and antibacterial properties of the hydrolysates.

### 3.5. Determination of Antioxidant Activity

The antioxidant activity was determined with three methods: by the ability to scavenge DPPH• and ABTS•+ free radicals and with the FRAP method. In every case, the activity of the lyophilized peptide mixture obtained after the two-step hydrolysis with the 1:10 E:S ratio was tested. The antioxidant measurements were repeated three times and the obtained values were averaged. 

The data indicated that the EYPe mixture had a significant antioxidant potential. It exhibited strong potency in scavenging DPPH• radicals and ABTS•+ radical cation decolorization. As reported by Memarpoor-Yazdi et al. [47] and Eckert et al. [48], the amino acid sequence of a peptide has a great impact on antioxidant activity. Such hydrophobic amino acids as A, V, L, P, M, Y, W, and F play a role in important antioxidant mechanisms. The aforementioned amino acids were abundant in the defatted egg yolk hydrolysates (Table 2). The analysis of the amino acid sequences of the peptides obtained after the hydrolysis showed a large amount of aromatic amino acids in many cases. Their aromatic residues stabilize radicals during the scavenging reaction through contributing protons to electron-deficient radicals [49]. Sheih et al. [50] suggest that the presence of Y can turn free radicals into phenoxy radicals, which leads to the inhibition of the peroxidation chain reaction.

The results of the DPPH• and ABTS•+ assays are in many cases higher than those determined for common dietary products. For example, the activity in the ABTS•+ decolorization assay reached 390.43 ± 8.92 mmol Trolox/kg (Table 7). It was higher than for such scavengers as spinach (8.49 mmol Trolox/kg), chili peppers (7.62 mmol Trolox/kg), or raspberries (16.79 mmol Trolox/kg) in dry mass [51]. Also, the DPPH• scavenging activity in the case of the EYPe mixture was much higher. As shown by the literature to date, strawberries are characterized by one of the highest ascorbic acid equivalents (520.70 mg AA/100 g) [31]. This result is much lower than that obtained for EYPe (1776.66 ± 32.99mg AA/100 g) (Table 7). The comparison of the results presented in Table 7 with those for egg yolk peptide fractions [48] showed that the separated and clear peptide fractions had higher values: from 0.50 to 2.75 µmol Trolox/mg in the DPPH• assay and from 19.90 to 88.25 µg Fe^2+^/mg in the FRAP test. This may be related to the high exposure to pure concentrated peptides. Our assumption was to create a mixture with antioxidant values in a simple way. Eckert et al. [43] showed comparable results of the free radical scavenging ability of phosvitin (0.11 μM Trolox/mg) after 5 h hydrolysis with bacterial proteases. Graszkiewicz et al. [45] demonstrated similar levels of free radical scavenging activity of egg protein hydrolysate (with trypsin) after separation on RP-HPLC. The values of 0.18 and 0.19 μM Trolox/mg were reached for two fractions.

A much weaker activity was calculated for ferric reduction. The peptide mixture exhibited a much lower FRAP activity than red cabbage (51.53 mmol Fe^2+^/kg) or red currant (44.86 mmol Fe^2+^/kg). This may be related to the small content of serine in the proposed peptides, whose hydroxyl group takes part in the reduction of metal ions. The significant differences in the values obtained with the free radical scavenging method and the FRAP method may be also caused by the differences in the reaction conditions. As suggested by Zou et al. [52], the effectiveness of the antioxidant method yielding high values of Trolox or ascorbic acid equivalents is related to the different underlying mechanism between the evaluation assays even for the same composition. It has been shown that an appropriate selection of antioxidant assays is highly correlated not only with the amino acid sequence but also with optimal reaction conditions.

### 3.6. Antibacterial Activity

The mechanism of action of the antimicrobial peptides involves binding to the cell membrane through an electrostatic attraction between the cationic residues of the peptides and anionic components or local lipids of the outer membrane. The antimicrobial peptides act on both Gram (+) and Gram (−) bacteria [18]. The EYPe mixture at the concentration of 50 mg/mL inhibited the growth of the studied strains (Table 8). Strong inhibition was detected in Gram (+) bacteria, while Gram (−) bacteria were less peptide-sensitive. Smaller inhibition zones were observed at the concentration of 25 mg/mL. These properties result from the presence of the sequences DSYEHGGEP, HVDLDEVANKIA, and FEDPERQESSRKE, which were identified to be responsible for antibacterial activity (Table 6).

Similar results were observed by Pellegrini et al. [53], who found that Gram (+) *Bacillus* strains were highly susceptible to the action of peptides digested by trypsin and chymotrypsin. Simultaneously, they reported a weak bactericidal activity of the peptides against Gram (−) bacteria. The antimicrobial activity of egg yolk peptides was also described by Pimchan et al. [54]. They fractionated peptides from pepsin-hydrolyzed egg yolks and identified 13 peptides with antibacterial activity, which inhibited the growth of *B. cereus*, *Escherichia coli*, *Staphylococcus aureus*, and *Salmonella typhimurium.* The growth of these species of bacteria is also effectively inhibited by plant extracts, which (like peptides), appear to be a good alternative to antibiotics especially as natural agents to prevent and control food-poisoning diseases [55].

## 4. Conclusions

The findings of the present study suggest that the proposed egg yolk peptide mixture may exhibit a number of pro-health properties. The peptides can function as a natural antioxidant given the high values obtained in the DPPH (1776.66 ± 32.99 mg AA/100 g) and ABTS (390.43 ± 8.92 mmol Trolox/kg) assays, which are comparable with those for well-known nutritional compounds considered as antioxidants. The antimicrobial properties of the EYPe have also been confirmed (the greatest against Gram (+) bacteria—*B. cereus* 1 with the value of 20.0 ± 1.0 mm at the concentration of 50 mg/L), which means that the peptides, which were proven to be potentially effective, can be used as a natural alternative to preserve food stuff and avoid the health hazards posed by chemical antimicrobial agents. The analysis of the MALDI-TOF/MS spectra revealed potential antioxidant peptide sequences (YPWTQR, ITMIAPSAF, DSYEHGGEP, VVSGPYIVY, QQGVEQGTR, KPQMTEEQIK, LMSYMWSTSM, HVDLDEVANKIA, YINQMPQKSRE, PGVTYPHPGQDTSAG, FEDPERQESSRKE, PTDQKVGWGGEGQIQ, YIEAVNKVSPRAGQ), which should be thoroughly investigated in the future.

## Figures and Tables

**Figure 1 foods-12-03394-f001:**
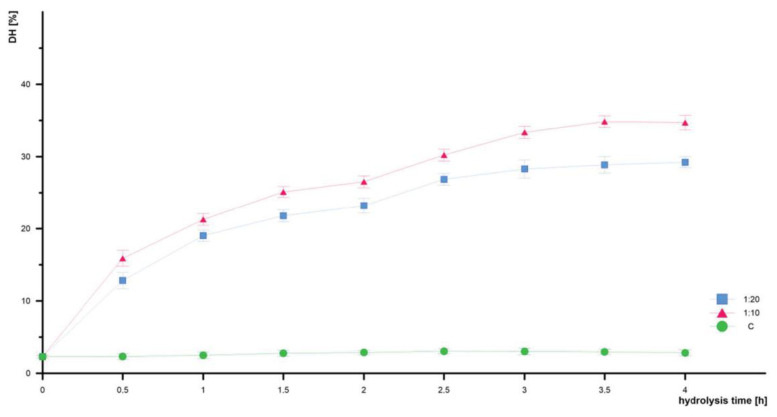
Effect of the E:S value on the degree of hydrolysis. C—control (without enzyme); results are shown from a 1:10 and 1:20 enzyme: protein ratio.

**Figure 2 foods-12-03394-f002:**
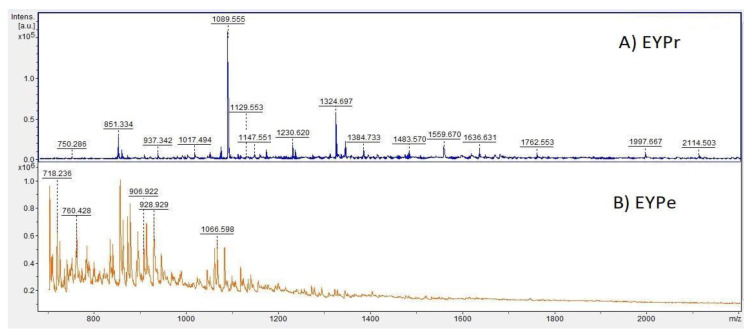
Mass spectra of (**A**) EYPr; (**B**) EYPe after two-step hydrolysis with a 1:10 E:S.

**Table 1 foods-12-03394-t001:** Average nutritional value in 100 g of liquid egg yolk (average ± standard deviation).

Energy Value	Fat	Carbohydrates	Proteins	Ash
1155 kJ/ 279 kcal	25.0 g ± 0.1	0.9 g ± 0.0	14.0 g ± 0.1	0.2 g ± 0.0

**Table 2 foods-12-03394-t002:** Amino acid composition of egg yolk protein hydrolysates and egg yolk protein (average ± standard deviation, mg/g protein, *p* < 0.05 mean statistically significant difference).

Amino Acid	Egg Yolk Protein (EYPr) [mg/g]	Hydrolysates (EYPe) [mg/g]	ANOVA *p* Value
Asp	54.3 ± 3.1	72.1 ± 4.1	0.000425
Thr	28.3 ± 1.9	38.5 ± 2.6	0.000728
Ser	48.4 ± 5.3	53.7 ± 5.9	0.223833
Glu	76.1 ± 4.2	103.4 ± 5.7	0.000240
Pro	23.3 ± 1.9	38.6 ± 3.2	0.000502
Gly	17.0 ± 0.1	33.9 ± 0.2	<0.000000
Ala	29.7 ± 0.7	40.5 ± 1.0	0.000006
Cys	7.9 ± 3.2	27.3 ± 11.0	0.033286
Val	32.4 ± 0.9	43.7 ± 1.3	0.000008
Met	15.5 ± 5.3	16.8 ± 5.7	0.748595
Ile	29.3 ± 0.9	35.9 ± 1.1	0.000079
Leu	50.7 ± 0.3	58.5 ± 0.3	<0.000000
Tyr	23.9 ± 2.2	32.5 ± 2.9	0.003252
Phe	25.6 ± 1.2	32.8 ± 1.6	0.000321
His	15.5 ± 2.2	18.3 ± 2.6	0.152067
Lys	41.9 ± 1.1	54.2 ± 1.4	0.000008
Arg	43.0 ± 1.2	51.4 ± 1.5	0.000146
Trp	11.4 ± 0.4	16.6 ± 0.6	0.000005

**Table 3 foods-12-03394-t003:** Lipid content (average ± standard deviation, %) in the dry matter of egg yolk.

Sample	EYPr	EYPe I Step	EYPe II Step	EYPe III Step	EYPe IV Step
Average [%]	50.6 ± 2.3	43.9 ± 1.3	35.5 ± 1.4	27.5 ± 1.5	18.9 ± 1.2

**Table 4 foods-12-03394-t004:** Effect of the E:S value on the hydrolysis degree (mean ± standard deviation, %).

E:S Ratio	Time of Hydrolysis [h]								
		Papain				Pepsin			
	0.0	0.5	1	1.5	2	2.5	3	3.5	4
**DH [%]**									
**Without enzyme**	2.30 ± 0.02	2.32 ± 0.04 *	2.49 ± 0.05 *	2.77 ± 0.04 *	2.87 ± 0.03	3.06 ± 0.04	3.02 ± 0.05	2.97 ± 0.04 *	2.84 ± 0.04
**1:20**	2.30 ± 0.02	12.82 ± 0.06 *	19.04 ± 0.04 *	21.78 ± 0.04 *	23.20 ± 0.05 *	26.87 ± 0.04 *	28.29 ± 0.17 *	28.87 ± 0.06 *	29.23 ± 0.04 *
**1:10**	2.30 ± 0.02	15.89 ± 0.06 *	21.27 ± 0.04 *	25.11 ± 0.04 *	26.53 ± 0.04 *	30.23 ± 0.04 *	33.36 ± 0.04 *	34.83 ± 0.04 *	34.71 ± 0.05 *

*—means statistically significant difference, *p* < 0.05.

**Table 5 foods-12-03394-t005:** Comparison of the DH value for different types of enzyme.

Enzyme	Digestion Time [h]	Condition	DH [%]	Source
Neutrase	3	200 U/mg, pH 7.0, 45 °C	12	[4]
Neutrase	2	Not available	27	[33]
Neutrase + Pronase	3—each enzyme	Not available	27	[43]
Alcalase + Protease N	3	E:S 0.5% (*w*/*w*)	25.3	[37]
Trypsin	3	Egg white, 1 U/1 mg protein, 30 °C	14	[45]
Trypsin	3	0.5% (*w*/*w*), 45 °C	15.6	[45]
Pepsin	2	Not available	45.3	[17]

**Table 6 foods-12-03394-t006:** Peptides obtained after papain–pepsin digestion with their activity checked in the BIOPEP database [46].

*m*/*z*	Sequence	Activity
849.381	YPWTQR	opioid ACE inhibitor
949.407	ITMIAPSAF	ACE inhibitor
990.481	DSYEHGGEP	antibacterial antioxidant
996.487	VVSGPYIVY	ACE inhibitor
1002.475	QQGVEQGTR	antiproliferative
1230.620	KPQMTEEQIK	antiproliferative
1236.595	LMSYMWSTSM	antioxidant
1324.697	HVDLDEVANKIA	antibacterial
1394.566	YINQMPQKSRE	ACE inhibitor
1483.570	PGVTYPHPGQDTSAG	antioxidant
1636.631	FEDPERQESSRKE	antibacterial antiproliferative
1670.622	PTDQKVGWGGEGQIQ	antioxidant
1681.584	YIEAVNKVSPRAGQ	ACE inhibitor

**Table 7 foods-12-03394-t007:** Antioxidant activity of the EYPe mixture sample (average ± standard deviation).

Sample	Ferric Reducing Ability (FRAP) [mmol Fe^2+^/kg]	Ferric Reducing Ability (FRAP) [µg Fe^2+^/mg]	DPPH Scavenging Activity [mg AA/100g]	DPPH Scavenging Activity [µmol Trolox/mg]	ABTS Radical Cation Decolorization Assay [mmol Trolox/kg]
EYPe	16.45 ± 0.19	0.11 ± 0.02	1776.66 ± 32.99	0.92 ± 0.04	390.43 ± 8.92

**Table 8 foods-12-03394-t008:** Antimicrobial activity of EYPe mixtures (average ± standard deviation, mm).

No.	Strain	Gram	Inhibition Zone [mm]	
			EYPe [50 mg/mL]	EYPe [25 mg/mL]
1.	*Bacillus cereus* 1	(+)	20.0 ± 1.0	10.7 ± 0.6
2.	*Bacillus cereus* 2	(+)	13.0 ± 1.0	9.0 ± 1.0
3.	*Bacillus megaterium*	(+)	13.7 ± 0.6	9.7 ± 0.6
4.	*Bacillus pumilus*	(+)	12.0 ± 1.0	9.3 ± 0.6
5.	*Kocuria rhizophila*	(+)	11.7 ± 0.6	7.7 ± 0.6
6.	*Serratia liquefaciens*	(−)	12.3 ± 0.6	8.3 ± 0.6
7.	*Pseudomonas aeruginosa*	(−)	9.3 ± 0.6	8.0 ± 1.0
8.	*Hafnia alvei*	(−)	9.3 ± 0.6	9.0 ± 1.0
9.	*Acinetobacter radioresistans*	(−)	8.7 ± 0.6	7.7 ± 0.6
10.	*Stenotrophomonas maltophila*	(−)	8.7 ± 0.6	7.7 ± 0.6

## Data Availability

The data used to support the findings of this study can be made available by the corresponding author upon request.

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
