# Peer review of "Egg Yolk as a New Source of Peptides with Antioxidant and Antimicrobial Properties"

_foods, 2023, doi:10.3390/foods12183394_

Round 1

Reviewer 1 Report

Comments and Suggestions for Authors

Comments to authors:

1.     Rewrite the abstract with quantitative results.

2.     Correct the language and flow of this sentence, “Resulting peptide mixture was tested 21 for its antioxidant properties by measuring scavenging activity on 2,2-diphenyl-1-picrylhydrazyl 22 radical (DPPH•), [2,2′-azinobis-(3-ethylbenzothiazoline-6-sulfonic acid) radical cation decoloriza-23 tion assay (ABTS•+) ferric reducing activity (FRAP) and also antimicrobial activity.”

3.     Introduction sections is nicely written to cover background. Besides this, language needs minor corrections.

4.     Give some information on egg yolk richness between country eggs (light brown) and poultry eggs (white).

5.     Line 81: provide the details of chicken from them eggs were obtained.

6.     Line 107: EtOH percentage given are in order used?

7.     Provide reference for protein determination.

8.     Section 2.2.8.1 and 2: what are the positive controls?

9.     Line 197: why 60 mins?

10Line 215-218: italicize the bacterial names.

11.  Table 2: recheck: the amino acids are in mg/gm?

12.  It will be nice to add in silico analysis (Uniprot/Expasy/ProtParam) results as peptide sequence is already available with authors.

13.  Line 332: instead of tree use three. Spell check the MS.

14.  For antioxidant activity discussion, it will be nice to compare results obtained with similar methods what authors used.

15.  Section 3.6: provide reason behind the bacterial growth inhibition or probable mechanism of action.

16.  Line 394-396: check the sentences for clarity and appropriateness.

Comments on the Quality of English Language

Need to improve.

Author Response

Thank you for your suggestions. I am sending the answers in the attachment.

Reviewer 2 Report

Comments and Suggestions for Authors

Dear author, please obtain the result from peer reviewing process in the attached document.

Thank you.

Comments on the Quality of English Language

Dear author, we are highly suggesting that this manuscript could undergo professional manuscript editing. Since the language contains many grammatical errors and ineffective sentences that make it difficult to be understood.

Author Response

(The authors gave the same response as above.)

Reviewer 3 Report

Comments and Suggestions for Authors

The authors conducted an investigation into the efficiency of delipidated egg yolk protein digestion, assessing the extent of hydrolysis across various enzyme-to-substrate ratios. Several comments have been raised that necessitate attention:

Comments:

1.     It is hoped that the authors will incorporate specific results into the abstract.

2.     Regarding lines 48 to 50, the statement "Hydrolysis mechanism allows to…" lacks clarity.

3.     At line 66, readers would benefit from a concise introduction to antimicrobial peptides and their modifications, as exemplified in Nat. Rev. Microbiol. 2020, 18 (5), 275-285. https://doi.org/10.1038/s41579-019-0288-0 and Chem. Soc. Rev., 2021, 50, 4932-4973 https://doi.org/10.1039/D0CS01026J.

4.     Within section 2.1, the process of isolating egg yolk and eliminating egg white proteins requires clarification.

5.     On line 96, typo of FeCl3∙6H2O; consistency is needed throughout.

6.     For Table 8, it is advisable to present the raw data of agar plates with antibacterial effects in the inhibition zone assay.

7.     In the antibacterial test, have any control groups involving conventional antibiotics been included?

8.     A recommendation is made for the authors to incorporate a discussion comparing antibacterial effects with those from alternative sources.

9.     The authors are urged to meticulously review chemical nomenclature and grammar for accuracy.

Comments on the Quality of English Language

see comments above

Author Response

Thank you for your suggestion. I am sending the answers in the attachment.

Round 2

Reviewer 1 Report

Comments and Suggestions for Authors

1.       Section 2.2.8.1 and 2: what are the positive controls? Here I am asking positive control with known potential to exhibit activity.

2.       It will be nice to add in silico analysis (Softwares: Uniprot/Expasy/ProtParam) results as peptide sequence is already available with authors. Check them online. Free to use. 

3.       Section 3.6: provide reason behind the bacterial growth inhibition or probable mechanism of action. This should be given with proper details.

Please be professional while answering the comments.

Comments on the Quality of English Language

NIL

Author Response

Thank you for your review. I am attaching the answers.

Reviewer 2 Report

Comments and Suggestions for Authors

Dear authors,

Thank you for your efforts that have been put during revision process. Hereby, I recommend this manuscript to be accepted as published article in Foods.
Congratulation and hope to see other great articles from you.

Thank you.

Author Response

Thank you for your review, insightful analysis and suggestions.

Reviewer 3 Report

Comments and Suggestions for Authors

The authors have addressed the main concerns.

For Table 8, it is advisable to present a few pictures of agar plates with antibacterial effects in the inhibition zone assay in the supporting information document.

Author Response

Thank you for your review.
